# Effect of Protein Denaturation Temperature on Rheological Properties of Baltic Herring *(Clupea harengus membras)* Muscle Tissue

**DOI:** 10.3390/foods10040829

**Published:** 2021-04-11

**Authors:** Agnieszka Strzelczak, Jerzy Balejko, Mariusz Szymczak, Agata Witczak

**Affiliations:** 1Department of Commodity Science and Quality Assessment, Process Engineering and Fundamentals of Human Nutrition, Faculty of Food Science and Fisheries, West Pomeranian University of Technology in Szczecin, 71-459 Szczecin, Poland; agnieszka.strzelczak@zut.edu.pl; 2Department of Toxicology, Dairy Technology and Food Storage, Faculty of Food Science and Fisheries, West Pomeranian University of Technology in Szczecin, 71-459 Szczecin, Poland; mariusz.szymczak@zut.edu.pl (M.S.); agata.witczak@zut.edu.pl (A.W.)

**Keywords:** fish proteins, denaturation, DSC, kinetics, rheology, Baltic herring

## Abstract

The technological properties of raw fish are influenced by the changes in protein structure under heating, which determines the texture and quality of the product. The aim of the study was to examine the protein denaturation temperature and the rheological properties of Baltic herring muscle tissue. The thermal properties were determined by the differential scanning calorimetry (DSC) method and the rheological properties were determined using dynamic oscillatory tests. DSC showed four peaks associated with denaturing transformations of myosin (39.59 °C), sarcoplasm (51.67 °C), connective tissue (63.16 °C), and actin (74.40 °C). Analysis showed that not all transformations occurred according to the same kinetic model. The first two and the last peak are described by 1st order kinetics, while peak 3 is described by 2nd order kinetics. Correlating the changes in fish tissue structure during heating with the rheological characteristics provides more information. The obtained kinetics models correlated very strongly with the results of model testing. Rheological changes of the G’ and G” values had two inflexion points and demonstrate a high degree of convergence with the DSC changes of herring muscle tissue from 20 to 85 °C.

## 1. Introduction

Thermal processing of raw fish material produces desired sensory properties, an improvement of raw material quality, and a safe product. Fish meat is subjected to thermal treatment in a wide range of temperatures from 30 to 200 °C during smoking (cold, warm, and hot), drying, cooking, steaming, sterilization in canning, frying, and baking. Changes in the protein structure that take place during heating have a significant effect on the technological and mechanical properties of fish raw material. Following thermal processing, the network of collagen fibres and proteins contracts, which is manifested by denaturation and aggregation, a change in the water holding capacity, and, consequently, a change in the fish structure into a harder or firmer texture. Another effect is the destruction of the cell membrane and the aggregation of sarcoplasmic proteins [1]. Heat treatment of meat also ensures food safety in production and distribution [2,3]. Fish muscle tissue contains approx. 70–84% water, 15–24% protein, 0.1–22% fat, 1–2% minerals, and 2% non-protein nitrogenous substances. The main types of muscle tissue proteins include myofibrillar, sarcoplasmic, and connective tissue (stroma) proteins [2]. Myofibrillar proteins (myosin and actin as an actomyosin complex, troponin, tropomyosin) are structural proteins that account for 65–70% of fish muscle protein. Fish myosin is unstable and very sensitive to denaturation, coagulation, degradation, and chemical changes. These proteins are responsible for water absorption, emulsification, and gelation. Sarcoplasmic proteins, which can be soluble in water and dissolved salt solutions, account for approx. 15–30% of the total protein content in fish muscles. These proteins are comprised of hundreds of enzymes, myoglobin, haemoglobin, and other albumins. Therefore, the presence of sarcoplasmic proteins has an adverse effect on the strength, deformability of myofibrillar protein gels, and water-holding capacity. In fish, stroma proteins consist mainly of gelatin and are located in the extracellular matrix, accounting for 3% of all muscle proteins. A low collagen content gives the fish meat a soft consistency [3,4].

The conformational changes in proteins that occur when they are heated are referred to as denaturation. The denaturation temperature for individual protein types is mainly determined using differential scanning calorimetry (DSC) [5,6,7]. The thermogravimetric and differential calorimetry tests applied to date are mainly concerned the thermal denaturation of the fish tissue to identify the temperature range for a particular transformation [8,9,10,11,12]. The kinetics of temperature-induced processes in fish meat are less frequently studied using TG-DSC measurements (thermogravimetric analysis and differential scanning calorimetry). These few studies mainly assessed the sequence of reactions, reaction rate, rate constant, pre-exponential factor, and the activation energy for a particular heating rate using the Arrhenius equation [13,14,15,16,17]. In turn, Santos et al. [18] and Sathivel et al. [19] applied the Coats–Redfern equation for the kinetic analysis of temperature-induced processes taking place in foodstuffs. These studies, however, did not address the fish muscle tissue. Therefore, the aim of the use of kinetic models in this study was to apply them to new food samples in order to predict at which temperature and how fast the denaturation of consecutive protein fractions would take place. This knowledge is applicable to optimization in the food industry, for mass yield, as well as to improvement of food quality measures, such as sensory assessment and texture.

Kinetic analysis causes difficulties, as kinetic models describe the theoretical simplification of the phenomena. Therefore, due to thermal equilibrium or mass transfer resistance in the gas phase, they may be insufficient to characterise the actual processes. Even less frequent studies concern the correlation of temperature-induced changes in the fish tissue structure with the rheological characteristics [18]. Therefore, the aim of this study was to combine a kinetic description of protein denaturation with a rheological description of texture changes, illustrated with an example of the herring, which is the most frequently processed fish in Europe.

## 2. Materials and Methods

### 2.1. Fish Raw Material

Baltic herring (*Clupea harengus membras* L.) were harvested for this study in January 2021. The fish were caught using nets within the area 27IIId of the Baltic Sea, in the Pomeranian Bay area. A batch of iced fresh fish was brought in 42 L boxes (Aquano, Fishbox H2) within 12 h of capture in accordance with PN-A-86761:1974 [20]. Prior to the study, ice was removed from the boxes and undersized or mechanically damaged individuals were discarded. Twelve herrings were randomly sampled and a morphometric analysis was conducted on them to determine the following: (i) quality index method (QIM) in accordance with Nielsen and Hyldig [21]; (ii) total fish weight with an accuracy of 0.1 g; (iii) total length with an accuracy of 1 mm; (iv) total weight of fillets and gonads with an accuracy of 0.1 g; (v) sex and sexual maturity according to Maier’s scale [22]; and (vi) gonad somatic index (GSI) and Fulton’s coefficient [23]. The water, protein, and lipids contents in the minced herring muscle tissue were determined in accordance with the AOAC (Association of Official Analytical Chemists) method [24]. 

For the differential scanning calorimetry (DSC) analysis, the fish were filleted manually and a sample was collected from the central part of the dorsal muscle using a lancet (*n* = 10). Samples for rheological testing were collected from a skinned fillet by cutting discs/cylinders with a 40 mm diameter and a 1.5 mm thickness using a special instrument.

### 2.2. Thermal Analysis 

Thermal properties were determined using the differential scanning calorimetry (DSC) method using a Q100 apparatus (TA Instruments, New Castle, DE). It was first calibrated for temperature and enthalpy using indium as a standard (T_m_: 156.6 °C; ΔH_m_: 28.45 J/g). The instrument was temperature-calibrated using water and indium. Enthalpy was calibrated with indium. Empty pans were used as a reference. The weight of the samples varied between 10 and 25 mg (accuracy of ± 0.01 mg). The measurements were carried out in one heating cycle in the temperature range from 5 to 100 °C and the rate was 5 °C/min. Denaturation temperature (T_d_) and denaturation enthalpy (ΔH) were estimated by measuring the area under the DSC transition curve.

### 2.3. Kinetics Analyses Based on DSC 

The rate of the denaturing process for proteins with the involvement of solid reactants was determined using Equation (1):(1)$$r=\frac{d\alpha }{dt}$$
where

*r*—reaction rate (1/min);*α*—degree of transformation;*t*—time (min).

To describe the theory of non-isothermal processes, including the relationship between the reaction rate and temperature, the Arrhenius equation was applied (which has the form of Equation (2) with a linear temperature change over time):(2)$$\frac{d\alpha }{dt}=\frac{A}{\beta }exp\left\{-\frac{E}{RT}\right\}f\left(\alpha \right)$$
where

*A*—Arrhenius equation coefficient (1/min);*E*—activation energy (J/mol);*R*—universal gas constant (J/mol×K);*β*—sample heating rate (K/min).

After integrating Equations (1) and (2), Equation (3) was obtained:(3)$$g\left(\alpha \right)=\frac{A}{\beta }\int_{T_{0}}^{T}exp\left\{-\frac{E}{RT}\right\}dT$$
where

*g*(*α*)—integral kinetic model of the process;*T*_0_—transformation onset temperature (K);*T*—transformation end temperature (K).

Protein denaturation processes proceed in several stages. Therefore, for each stage, both the form of the function *g*(*α*) (kinetic model) and the Arrhenius equation parameters A and E were determined. An analysis of the process under non-isothermal conditions was determined based on the effect of sample heating rate and temperature on the degree of transformation and the process rate. For the assumed transformation degree range, the *g*(*α*) function values were calculated by applying a formula appropriate for a particular kinetic model. After identifying models for particular stages and determining the A and E parameters, a complete description of the process was obtained. The results were used for an analysis of the course of the experimental process with the theoretical one, resulting from the neural model. 

The effect of sample heating rate and the temperature on the extent of reaction was identified by determining the function Equation (4) value from the Coats–Redfern equation: (4)$$ln\left(\frac{g\left(\alpha \right)}{T^{2}}\right)=F\left(T\right)$$
where

*F(T)*—random function for several measurement series.

The *g*(*α*) was then calculated in accordance with Equation (5):(5)$$g\left(\alpha \right)=T^{2}F\left(T\right)$$

The *g*(*α*) function value was calculated from the equation describing this function. The best fit was obtained for the F1 and F2 models, according to Hastings et al. [10] (Table 1).

### 2.4. Rheological Measurements

The AR 2000ex rheometer (TA Instruments, New Castle DE) equipped with manufacturer-supplied computer control software (Rheology Advantage Data Analysis Program) was used to study the dynamic oscillatory rheological behavior of the different parts of herring muscles. A 40 mm cross-hatched parallel plate made of stainless steel was used with a gap of 1500 µm. The AR 2000 was supplemented with an efficient Peltier temperature control system and the sample temperatures were precisely controlled and monitored. For each test, a measured whole tissue sample was placed on the bottom plate of the rheometer. The sample temperature was increased from 20 to 85 °C with incremental steps of 5 °C. The exposed sample perimeter was covered with a ceramic trap to minimise evaporation at higher temperatures. Dynamic oscillatory tests were carried out at a frequency of 1 and 10 Hz. 

All rheological measurements were carried out in triplicate. The storage modulus (G”, a measure of elastic property), loss modulus (G”, a measure of viscous property), and the phase shift angle (δ) were obtained directly from the software (Rheology Advantage, TA Instruments—Waters LLC TRIOS version 5.1.0.46403).

### 2.5. Statistical Analysis

Significant differences (*p* < 0.05) between the average values were identified using Tukey’s test. This study assessed the model identification and the deviation from linearity by the neural network method. The learning algorithm BFGS Broyden–Fletcher–Goldfarb–Shanno [25,26] was applied and implemented in the Statistica 13 program (Statsoft, Tulsa, OK, USA, http://statistica.io; accessed on 01 October 2020) with an artificial neural network module (TIBCO Software Inc., Palo Alto, CA, USA, 2017). 

## 3. Results and Discussion 

The Baltic herring harvested in January and delivered to the laboratory received eight points in the QIM analysis, which indicates the 3rd day of storage in ice following the catch. The fish were in the *post rigor-mortis* condition. The average weight of the fish was 170 ± 21 g, the length was 249 ± 2.3 cm, the weight of fillets was 93 ± 12 g, and the weight of gonads was 32 ± 5 g. The herrings were of grade VI of sexual maturity, the GSI was 17.5, and Fulton’s index was 1.12. The herring muscle tissue contained 75.8 ± 0.3% water, 8.7 ± 0.2% lipids, and 16.2 ± 0.15 g protein. The results show that the tested herring was in the season preceding entering of the spawning grounds. This means that the tissue of the tested herring was tender, with the lower collagen content and changes in myofibrillar proteins that occur in a lean herring after spawning [27,28]. Kołakowski et al. [29] showed that the bream fishing season had no significant effect on the protein fraction coagulation temperature. Nevertheless, the sarcoplasmic protein peaks for the bream during the spawning season were smaller in size, which was explained by the lower solubility of muscle proteins. In turn, it was found for the hake that the proteins of fish in a better biological condition (post-spawning) were denatured more quickly and completely than during the pre-spawning period [30]. 

Analysis of the DSC curves showed four peaks associated with subsequent denaturation transformation of four protein fractions in the tested Baltic herring meat samples (Figure 1, Table 2). The peaks corresponded to the following fractions: 1—myosin, 2—sarcoplasm, 3—connective tissue, and 4—actin [31]. Average denaturation temperatures (T_d_) for subsequent peaks were determined to be 43.45 °C, 56.79 °C, 69.01 °C, and 77.91 °C, respectively (Table 2). Similar denaturation temperature ranges for herring proteins were obtained by Hastings et al. [10], and others [32]. Gill et al. [33] also showed that an increasing salt concentration had a minor effect on the herring myosin denaturation temperature. Four peaks are visible on each of the DSC thermograms, however, they probably overlap and it is not possible to accurately read the initial and final denaturation temperatures of each protein fraction. On the other hand, the results clearly show that the herring protein begins to denature at 39.65 ± 1.58 °C and ends at 81.62 ± 1.22 °C. 

Kinetic analysis demonstrated that the two former peaks and the latter peak are described using an F1 model (kinetics of the 1st order) and peak 3 using an F2 model (kinetics of the 2nd order). This means that the protein denaturation processes for peaks 1, 2, and 4 proceed in accordance with the linear temperature dependence, in contrast to the protein denaturation for peak 3 (Table 1). Neural network-based statistical analysis identified these differences and identified the model with the highest degree of fit, as indicated by the high value of the Spearman’s correlation coefficient (Figure 2, Table 3). The developed theoretical kinetic model exhibits a very strong correlation with the results obtained during the study. This means that the obtained kinetic model can be used in the future in the fish industry to optimize the dynamics of protein denaturation during herring processing under different technological conditions. Using the identified model, we can predict the output (conversion degree) for new samples using temperature and *g*(*α*) as input variables in the elaborated artificial neural network ANN models. 

For the tested herring batch, the value of standard deviation (SD) of denaturation temperature amounted to several per cent, which indicates that repeatability of the analysis was high. In turn, great differences were obtained for the E activation energy of the Arrhenius equation coefficient. Such great differences in results for individual fish in the bath may be due to the wide range of water and lipid contents. As for the herring, the lipid content in the muscle tissue of different fish in a batch may differ from two to five times, depending on the catching season. Therefore, a batch of herring obtained from fisherman before the DSC study was carefully divided into homogeneous and representative fish from atypical and damaged fish. Stodolnik and Gabryszak [34] showed that triacylglycerols could protect fish proteins against thermal denaturation. This may also be due to the different rates of interactions between post-mortem proteins in fish. The results also show a reduction in the enthalpy required to induce protein denaturation (data not shown) with no significant changes in T_d_, which may indicate progress in the herring muscle tissue autoproteolysis [35]. The tested herring in degree VI of gonad maturity exhibit the highest activity of endogenous peptidases throughout the annual season. The DSC analysis demonstrated no small endothermal peaks below 40 °C, which are an indicator of significant proteolytic autolysis [36]. Schubring and Meyer [32] showed that herring muscle proteins were not denatured during the storage of fish in ice. Therefore, the differences in temperature may be due to various pH of the test material and the intramuscular connective tissue content, which have a significant effect on the shift of protein denaturation temperatures [37].

The rheology of fish products is directly related to thermal denaturation and gelation of protein fractions. Rheology plays an important role in product development, sensory acceptance, quality control, and device design. Complex rheological behavior is an effect of the action of a single protein or multiple proteins. The results show that the initial thickening/gelling temperature in the Baltic herring is approx. 40 °C, and is related to a complex dynamic rheological behavior. Denatured proteins can mutually interact, thus leading to the aggregation and formation of gel.

Storage modulus G’ provides information about the elastic measure of herring meat. Changes in the G’ of the herring meat during the heating from 20 to 85 °C can be divided into three separate series (S1, S2, and S3) using two inflexion points (IP1 and IP2), where the first inflexion point is characteristic of the onset of muscle protein denaturation, and the second inflexion point for the completion of protein denaturation (Figure 3).

The G’ value in Series 1 clearly decreases within the range of 20–40 °C, and after the IP1 has been exceeded, it also clearly increases while crossing the denaturation points for individual proteins. The increase in G’ during the heating segment, Series S2, was attributed to the development of a thermohardening gel matrix produced by the herring meat [38]. It was additionally strengthened as the recorded IP1 value (~40 °C) reflects the typical denaturation temperature for the herring myosin and the peak observed in the DSC data. The bend of the G’ line in the central part of the S2 series, marked as T_gel_ point, was observed within the range of 52–54 °C, which indicates the gelation temperature for the Baltic herring meat (Figure 3). This means that the transition from the viscoelastic behavior to elastically-viscous is 53 °C for the herring muscle tissue. The behavior transition of herring meat at 53 °C is mainly prompted by total myosin denaturation and denaturation of some proteins from the sarcoplasmic fraction (Table 3). The G’ value increased clearly up to a temperature of approx. 70 °C (Figure 3), which shows that the denaturation of sarcoplasmic and stroma proteins contributes to an increase in the gel hardness. Within the range of 70–75 °C, the G’ value is stable, which indicates that the thermally-induced denaturation and aggregation of a protein mixture ended at that temperature. This means also that denaturation of actin proteins had a smaller ratio in gel hardness than that of other proteins fractions. A further increase in the system temperature up to 85 °C resulted in a reduction in G’, which can be explained by the thermohydrolysis of proteins, in particular collagen [39]. In addition to proteins, herring tissue also contains water and lipids. These liquids in contrast to proteins are inelastic, that means they are viscous. Loss modulus (G’’) provides information about the viscosity of herring meat. Increasing the meat temperature from 20 to 40 °C reduced the viscous modulus (G”) from 2.6 to 1.8 kPa—this is almost 30% (Figure 3). Denaturation of myosin and sarcoplasmic proteins slowed the reduction of the viscous modulus, which decreased by only 0.1 kPa. In contrast, an increase in temperature from 56 to 70 °C combined with denaturation of connective tissue and sarcoplasmic proteins increased the viscous modulus of the meat by 0.2 × 10^−3^. Heating the meat above 70 °C caused another rapid decrease in the viscous modulus of the meat. Storage modules (G’—elastic) and loss modulus (G”—viscous modulus) create a complex modulus (G*), which shows the overall resistance to deformation of herring meat. The lower value of G” than that of G’ has also been reported for many other marine fish species [40]. The angle of the G* value slope is called the delta (δ), which indicates the proportion of viscosity to elasticity of the meat. The values of the initial phase shift (δ, in degrees) were also recorded for the herring meat (Figure 3). The delta (δ) changed the value at 40 °C from 12.3° to 10–11°, and at 70 °C to 7°, which indicates a change in the gel structure from very elastic to elastic. Lower δ values represent formation of a better three-dimensional network [41]. It can be observed that the delta decreases by a few degrees Celsius earlier than the onset of the denaturation of the IP1 herring myosin (Figure 3). No such effect was observed for IP2 at a temperature of approx. 70 °C. This may confirm that the aggregation of protein precedes its gelation and is related to different interactions of a part of the myosin tail and head regions [42]. Chan et al. [43] indicated that the aggregation rate of fish myosin through non-covalent bonds increased rapidly at temperatures between 32 and 50 °C. These phenomena were explained by the gelation kinetics, in which aggregation is favoured at slow heating [44], which were used in the DSC analysis and rheology in this study.

Heating has a significant effect on the rheological properties of fish meat. It was found that actin denaturation resulted in an increase in firmness and decreased juiciness, while the denaturation of myosin and collagen was responsible for increased firmness and decreased solidity of the muscle fibres. In general, thermal denaturation induces the contraction of the muscle fibres, which is perceived as the hardening of meat. In turn, collagen heated in a humid environment is transformed into soluble gelatin, which decreases the meat hardness. Additionally, having analysed the denaturation temperature ranges for individual protein types, it is possible to optimally select the thermal processing parameters for the fish raw material type and, thus, to have a direct effect on the final quality of the product, its stability, and functionality. 

## 4. Conclusions

The thermodynamic properties of fish raw material proteins have an effect on the production methods and quality of the finished products. The DSC analysis of the Baltic herring muscle tissue demonstrated that the proteins of Baltic herring caught in January 2021 had a denaturation temperature ranging from 31.6 to 81.7 °C. Myosin denatures at the lowest temperature, followed by sarcoplasmic proteins and connective tissue proteins, while actin denatures at the highest temperature. The statistical analysis also showed that most herring proteins denature in accordance with the straight-line function, depending on the temperature. The kinetic denaturation models obtained in the statistical analysis correlate very strongly with the results of the model testing. The denaturation of proteins results in their aggregation and gelation. Changes in the G’ and G” values show a high degree of convergence with denaturing changes in the tested samples of herring meat. Rheological analysis demonstrated that protein gelation resulted in an increase in the storage modulus (elastic of meat) and a decrease in the phase shift angle (viscosity-to-elasticity ratio) in the temperature range from 20 to 85 °C, with a simultaneous decrease in the loss module value. This indicates an increase in the degree of elasticity, which is a consequence of strengthening of the tissue structure by the denaturation of subsequent protein fractions.

## Figures and Tables

**Figure 1 foods-10-00829-f001:**
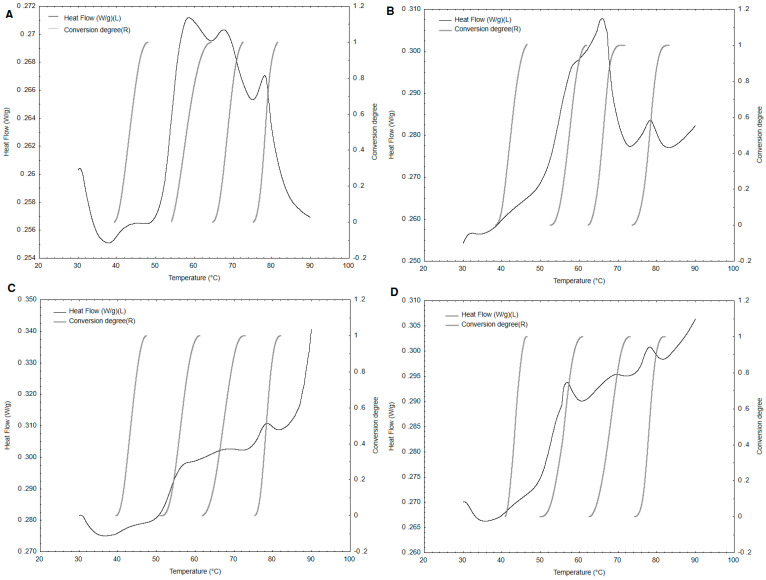
Conversion degrees calculated for consecutive stages from the differential scanning calorimetry (DSC) of Baltic herring (*n* = 4, consecutive samples (**A**–**D**)).

**Figure 2 foods-10-00829-f002:**
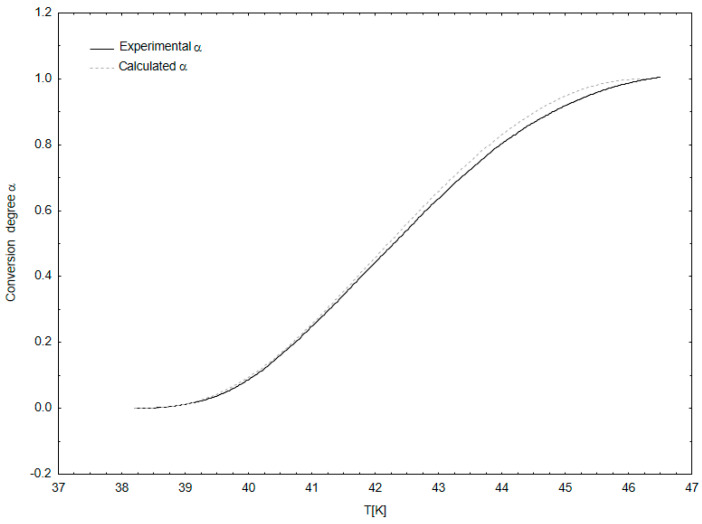
Comparison of experimental *α*(T) functions and those calculated from the kinetic model for stage I of protein denaturation.

**Figure 3 foods-10-00829-f003:**
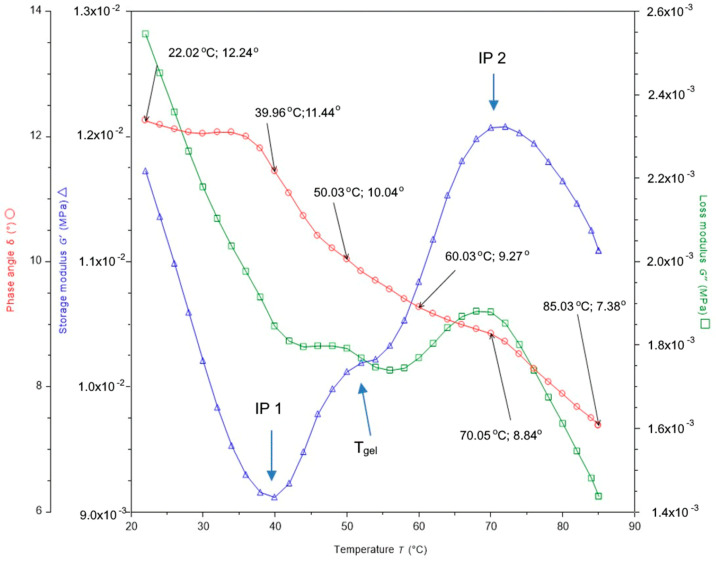
Changes of storage modulus (G’), loss modulus (G”), and phase shift angle (δ) during heating of herring meat.

**Table 1 foods-10-00829-t001:** List of kinetic models [10].

Mechanism	Symbol	g(a)
1st order kinetics	F1	[−ln (1 − *a*)]
2nd order kinetics	F2	(1 − *a*)^−1^ − 1

**Table 2 foods-10-00829-t002:** Results of DSC analysis of Baltic herring muscle tissue (*n* = 4).

Sample	Peak Number	T_d_ (°C)	Kinetic Model	E (J/mol)	A (1/min)
A	1	44.50	F1	244.46	1.30 × 10^16^
	2	56.94	F1	125.36	1.69 × 10^8^
	3	72.29	F2	116.67	3.04 × 10^7^
	4	76.84	F1	183.65	3.71 × 10^11^
B	1	43.28	F1	92.53	5.25 × 10^5^
	2	57.85	F1	86.43	1.48 × 10^5^
	3	69.09	F2	191.51	1.43 × 10^12^
	4	78.79	F1	97.11	1.19 × 10^6^
C	1	42.35	F1	90.33	2.93 × 10^5^
	2	57.13	F1	223.93	4.13 × 10^14^
	3	66.91	F2	115.22	3.04 × 10^7^
	4	77.87	F1	106.42	5.23 × 10^6^
D	1	43.66	F1	183.60	3.70 × 10^11^
	2	55.26	F1	92.72	5.48 × 10^5^
	3	67.76	F2	86.40	1.47 × 10^5^
	4	78.13	F1	245.16	1.20 × 10^16^
Average from A–D samples	1	43.45 ± 0.89	F1	152.7 ± 375.02	1.23 × 10^11^ ± 2.14 × 10^11^
	2	56.79 ± 1.10	F1	133.36 ± 62.56	1.03 × 10^14^ ± 2.06 × 10^14^
	3	69.01 ± 2.36	F2	137.45 ± 45.74	3.58 × 10^11^ ± 7.15 × 10^11^
	4	77.91 ± 0.81	F1	158.09 ± 69.82	3.00 × 10^15^ ± 6.00 × 10^15^

**Table 3 foods-10-00829-t003:** Quality of neural models with Spearman’s rank correlation coefficients between experimental and calculated data. MLP (Multi Layer Perceptron) input neuron/hidden neurons/output neurons.

Sample	Peak Number	Neural Model	Spearman’s R Correlation Coefficient
A	1	MLP 1:24:1	0.998
	2	MLP 1:32:1	0.989
	3	MLP 1:36:1	0.973
	4	MLP 1:24:1	0.998
B	1	MLP 1:45:1	0.976
	2	MLP 1:31:1	0.969
	3	MLP 1:32:1	0.983
	4	MLP 1:24:1	0.958
C	1	MLP 1:21:1	0.979
	2	MLP 1:29:1	0.972
	3	MLP 1:53:1	0.993
	4	MLP 1:25:1	0.981
D	1	MLP 1:21:1	0.989
	2	MLP 1:25:1	0.999
	3	MLP 1:24:1	0.988
	4	MLP 1:34:1	0.978

## Data Availability

The data presented in this study are available on request from the corresponding author.

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
