# Peer review of "Effect of Protein Denaturation Temperature on Rheological Properties of Baltic Herring (Clupea harengus membras) Muscle Tissue"

_foods, 2021, doi:10.3390/foods10040829_

Round 1
Reviewer 1 Report
The authors did not change the analysis of the thermal analysis results. I still think that the analysis of the thermograms is wrong because they consist of 4 partially overlapping peaks. Therefore, it is not possible to define, for each of them, the onset and endset temperatures and the corresponding enthalpy. In addition, the thermograms of the four samples are very different and not very reproducible.
The authors have reviewed the presentation and discussion of rheological test results, but there are some evident errors about basic rheological concepts:
they report a transition from the visco-elastic behaviour to the elastically-viscous one. What does it mean?
they confuse the viscous modulus with the viscosity...
I suggest rewriting it.
Author Response
the answer is included in the appendix.

Reviewer 2 Report
The revisions were made based on the reviewer's comments.
Author Response
The answer is included in the appendix.

Round 2
Reviewer 1 Report
The authors replied satisfactorily to my comments.
This manuscript is a resubmission of an earlier submission. The following is a list of the peer review reports and author responses from that submission.
Round 1
Reviewer 1 Report
This paper studied the effects of heating of fish muscle tissue on the thermal denaturation of proteins and rheological behavior. The fish muscle tissues were analyzed by DSC and rheometer during heating. The conversion degrees of four different protein fractions obtained by DSC were fitted to different kinetic models.
The reviewer recommends that the manuscript’s acceptance should be pending until some revisions will be made.
Typographical errors found throughout the manuscript (e.g., lines 29, 38, and 59).
The purpose to use the kinetic models was unclear.
The relationship between the results from DSC and rheological analysis lacked.
Reviewer 2 Report
In this manuscript, the authors utilized differential scanning calorimetry to monitor the rheological change during protein denaturation. I have some minor comments as follows. Comments: 1. Page 1, line 21: the full name of DSC is missing.
2. Page 1, line 29: there should be a space between structure and taking.
3. Page 1, line 35; page 2, lines 50 and 58: correct the indents.
4. Page 1, line 38: the symbol [ before Myofibrillar should be removed.
5. Page 7&8, Figure 1: Can authors rearrange 4 small figures on the same page?
6. Page 9, Figure 2: the resolution is low, please improve.
7. Page 12, line 429: methodology, AS i JB. Should i be replaced by comma?
8. Page 12, line 440: the dash between physio and chemical should be removed.
9. Page 13, line 459: the species name should be italic.
10. Page 13, line 472: “85, 51-58 (2), 419-428” is weird, please fix.
Reviewer 3 Report
The aim of the manuscript is not clearly definite by the authors (what they reported in the abstract is different from what is reported in the introduction). The experimental design has a strong weakness: only 12 samples were analyzed despite the fact that the authors state that there is a strong variability between herring samples. The manuscript is not clear and not well organized, its accuracy is fair: some methods are reported (content of water, protein, lipids) but the corresponding results are not presented; the meaning of several sentences is not clear; some references are not appropriate (as 13-17); the equation 2 is not correct. Moreover, in my opinion the presentation, the interpretation and the discussion of the results should be completely revised. For what concern the DSC results, each thermogram cannot be considered as constituted of four separate peaks but as a single peak with relative and absolute maxima. As a consequence, the data analysis is wrong. I disagree about the discussion of the viscoelastic behavior of the sample during heating. Considering that herring fillet is solid, how can the authors refer to a transition from a fluid-like behavior to a solid one? The protein transitions cannot be discussed as a gel transition with a subsequent increase of the gel hardness. Moreover, it is not clear how modelling the kinetics of protein denaturation helps to understand the rheological behaviour of herring fillets during heating.